# Added value of an open narrative in verbal autopsies: a mixed-methods evaluation from Malawi

Patricia Loh,[1] Edward Fottrell,[1] James Beard,[1] Naor Bar-Zeev,[2,3] Tambosi Phiri,[4] Masford Banda,[4] Charles Makwenda,[5] Jon Bird,[6] Carina King  [1,7]

► Additional material is published online only. To view please visit the journal online (http://dx.doi.org/10.1136/bmjpo-2020-000961).

[1]Institute for Global Health, University College London, London, UK
[2]Malawi-Liverpool-Wellcome Trust Clinical Research Programme, Blantyre, Malawi
[3]International Vaccine Access Center, Department of International Health, Bloomberg School of Public Health Johns Hopkins University, Baltimore, Maryland, USA
[4]MaiMwana Project, Mchinji, Malawi
[5]Parent and Child Health Initiative, Lilongwe, Malawi
[6]Department of Computer Science, University of Bristol, Bristol, UK
[7]Department of Global Public Health, Karolinska Institute, Stockholm, Sweden

**Correspondence to**
Dr Carina King; c.king@ucl.ac.uk

## ABSTRACT

**Background** The WHO standardised verbal autopsy (VA) instrument includes closed questions, ascertaining signs and symptoms of illness preceding death, and an optional open narrative. As VA analyses increasingly use automated algorithms, inclusion of narratives should be justified. We evaluated the role of open narratives on VA processes, data quality and respondent's emotional stress.

**Methods** A mixed-methods analysis was conducted using VA data for child deaths (0–59 months), between April 2013 and November 2016 in Mchinji district, Malawi. Deaths were prospectively randomised to receive closed questions only or open narrative followed by closed questions. On concluding the VA, interviewers self-completed questions on respondents' emotional stress. Logistic regression was used to determine associations with visible emotional distress during VAs. A group discussion with interviewers was conducted at the project end, to understand field experiences and explore future recommendations; data were coded using deductive themes.

**Results** 2509 VAs were included, with 49.8% (n=1341) randomised to open narratives. Narratives lasted a median of 7 minuntes (range: 1–113). Interviewers described improved rapport and felt narratives improved data quality, although there was no difference in the proportion of deaths with an indeterminate cause using an automated algorithm (5.3% vs 6.1%). The majority of respondents did not display visible emotional stress (81%). Those with a narrative had higher, but not statistically significant, odds of emotional distress (adjusted OR: 1.20; 95% CI: 0.98 to 1.47). Factors associated with emotional stress were: infant deaths versus neonates; deaths at a health centre or en-route to hospital versus home; and higher socioeconomic status. Non-parental respondents and increased time between death and interview were associated with lower odds of emotional distress.

**Conclusion** Conducting an open narrative may help build rapport, something valued by the interviewers. However, additional time and emotional burdens should be further justified, with quality and utility of narratives promoted through standardised recommendations.

## BACKGROUND

A comprehensive civil registration and vital statistics (CRVS) system recording births and deaths provides a country with essential information to make informed decisions for country-specific priority setting, and measure its progress towards the Sustainable Development Goals.[1 2] In the absence of functional CRVS structures, verbal autopsies (VAs) can partly fill this gap as an interim mortality data collection instrument by providing cause-specific mortality estimates.[3–5] These data can be used by governments, healthcare providers, researchers, donors and policymakers, who rely on accurate and comparable data over time to estimate burdens of diseases at population level, evaluate programme implementation and complement routine administrative data.[1 6] A recent review supported the use of VA to identify vulnerable groups and

### What is known about the subject?

► Verbal autopsies are often conducted in contexts where civil registration systems are lacking or incomplete.
► Several tools for conducting VAs exist - some containing an open narrative section, where respondents describe the events leading to a death in their own words.
► Automated methods for analysing VAs often don't use data from narratives, therefore there should be a clear and justified reason for conducting this interview section .

### What this study adds?

► Data collectors reported the narratives as a way to build rapport with the respondents and felt this improved their ability to collect quality information.
► While respondents mostly did not show visible signs of emotional stress during interviews, this was more frequent but statistically non-significant, in those with a narrative.
► There may be a trade-off in the increased time and emotional burden of VAs withnarratives, with the ability to establish a connection between data collectors and respondents.

health needs for effective resource allocation in humanitarian settings.[7]

The VA process involves trained fieldworkers identifying and interviewing an appropriate respondent, usually a close relative or caregiver, for a given death. Events preceding the death are recorded using a survey with a predetermined set of closed questions, which can be supplemented by a free-text open narrative designed to elicit the story in the respondent's own words of how the death occurred.[8] Following this, a suspected cause of death is generally assigned through physician review or through the automated application of statistical algorithms (eg, InterVA or SmartVA).[3–5]

In 2006, up to 18 VA tools with varying combinations of closed questions and open narratives were reportedly being used in 13 countries.[9] The WHO published the first iteration of a standardised VA methodology in 2007, with subsequent updates in 2012, 2014 and 2016.[8] The inclusion of an open narrative section remains recommended, but optional. The role of the narrative in physician-coded VAs has been likened to a medical history used by doctors to make diagnoses.[10] It can also encourage interviewer–respondent rapport, providing respondents a more natural outlet to express themselves and recount events they feel were most relevant.[11] The open narrative can also provide valuable information that standardised closed questions do not capture, such as cultural beliefs, adding context and holding authorities accountable to design interventions and services that are responsive to its people's needs.[9 12 13] In contrast, it could be argued that such information could be better identified using structured social autopsy tools—a supplementary survey conducted specifically to identify non-medical causes of death.[7 14]

The emotional strain of a VA has been detailed in qualitative studies from Ghana,[15] Papua New Guinea[16] and Nepal,[17] and fieldworkers from South Africa reported a higher likelihood of respondents becoming emotionally stressed during the open narrative compared with closed question sections of the interview.[18] Furthermore, the potential for adverse effects of VA-induced distress on data quality, and the diagnostic influence this might have for assigning cause of death is important to understand.[12 19 20]

This paper explores the role of the open narrative in the VA interview process, including its effects on procedures, data quality and emotional stress in respondents. Narratives potentially pose additional burdens on both respondents and interviewers, and as VAs are increasingly analysed using automated algorithms that do not use these free-text responses, their inclusion in the VA process should be justified.

## METHODS

We conducted a mixed-methods analysis of VA process data for deaths of children aged 0–59 months collected prospectively from April 2013 to November 2016, as part of the VacSurv Study in Mchinji district, Malawi.[21] Mchinji is a rural agricultural district in the central region of Malawi, with a population of approximately 600 000, under-5 mortality rate of 63 per 1000 live births and crude birth rate of 32.2 at the time of data collection.[22]

### Mortality surveillance

Full details of the population surveillance system used by the VacSurv Study have been previously described.[21 23] Briefly, deaths in children aged 0–59 months, including stillbirths, were registered retrospectively from October 2011 to February 2012, and prospectively from March 2012 to June 2016. Births and deaths were reported by 1060 volunteer village informants who cumulatively covered the whole of Mchinji district, supervised by 50 enumerators and 8 senior monitoring and evaluation officers (MEOs). Data were submitted using paper forms to the central office monthly where it was entered into a Microsoft Access database. Major errors in identification data (eg, incompatible dates of birth and death) were sent back to the field for verification. All deaths in children under-5 years were extracted from the cleaned data, and preprinted forms with a unique barcode containing the participant's study ID were generated.

### Verbal autopsies

Deaths were prospectively randomised to one of two standard VA approaches: (1) closed questions only or (2) open narrative followed by closed questions. Randomisation was programmed into the electronic data capture form (Open Data Kit software),[24] and the MEOs were informed of the allocation after the respondent had consented to the interview. The respondent was blinded to the randomisation procedure to minimise potential recall and volunteer biases, but MEOs were unblinded to the purpose of randomisation. The open narrative was unstructured and MEOs could choose how they recorded the details, such as audio-recording and subsequent transcription, notes or direct transcription of the story during the interview. The closed questions used WHO's 2012 VA instrument,[25] translated into Chichewa.

### Data collection

The VAs were conducted at respondents' homes by nine different MEOs. The MEOs were all Malawian men, who resided in Mchinji district and had completed secondary education. Several hold diplomas in community mobilisation and social work. All had worked within the local communities for a minimum of 5 years before project commencement and had conducted VAs previously. They underwent a 1-week training, including: collective translation of the WHO VA questionnaire; study protocol including data collection using smartphones; developed standard operating procedures (SOPs) for conducting the interviews sensitively and supervised mock interviews.[12] The SOPs included identifying the main respondent, consent procedures, managing respondent distress and offering condolences.

At the end of each VA, MEOs self-completed post-interview questions. MEOs were asked to document the respondents present, emotional stress during the interview and whether the interview needed to be paused as a result. Total VA interview duration was automatically captured on the smartphone, and MEOs noted the start and end time of the open narrative on the paper form.

Closed questions were collected using ODK Collect on Android smartphones and narratives were submitted as written transcripts on the preprinted forms. These were entered into a Microsoft Access database, and data were linked using the participants' study ID, then cleaned and processed.

### Quantitative analysis

Child characteristics and VA process data were described with proportions, means and SDs for normally distributed continuous data or medians and IQRs for asymmetrically distributed data. Student's t-test and $X^2$ test were applied to the comparisons of process data between those with and without narratives.

Cause of death was assigned using InterVA-4 (www.interva.net) based on closed question responses only; respondents had the option of answering with 'yes', 'no' or 'don't know'. InterVA uses a Bayesian model to assign the posterior probability of different causes of death based on positive ('yes') closed question responses only. The number of 'yes' answers and subsequently the ability to assign probable cause of death were used as a proxy measure of data quality. Emotional distress and interview duration were chosen as proxy indicators of burden for respondents and interviewers. Stillbirths were excluded from the analysis as we used a locally modified VA tool for these deaths. The narratives were not used for assigning cause of death, and it was outside the scope of the study to validate or verify automated cause of death assignment.

The primary analysis was a per-protocol analysis (ie, excluding interviews not conducted as allocated). This was chosen to examine the mechanism of narratives and the relationship to respondent distress, and not the process of recommending a narrative be done. We compared respondent emotional distress during the interview between those with and without a narrative. A multivariable logistic regression was conducted, adjusted for potential confounders defined a priori as: main respondent, child's age and sex, location of death and socioeconomic tercile. All analyses were conducted with Stata V.15.0.

### Qualitative data collection and analysis

At study completion, a group discussion was held with the MEOs who conducted VAs during the project to gather their feedback on the utility of the open narrative, their recommendations for future VA procedures and debrief on the emotional toll of administering VAs. This group discussion was led by the technical advisor (CK) in a private room within the office using a structured topic guide (online supplemental appendix 1). The

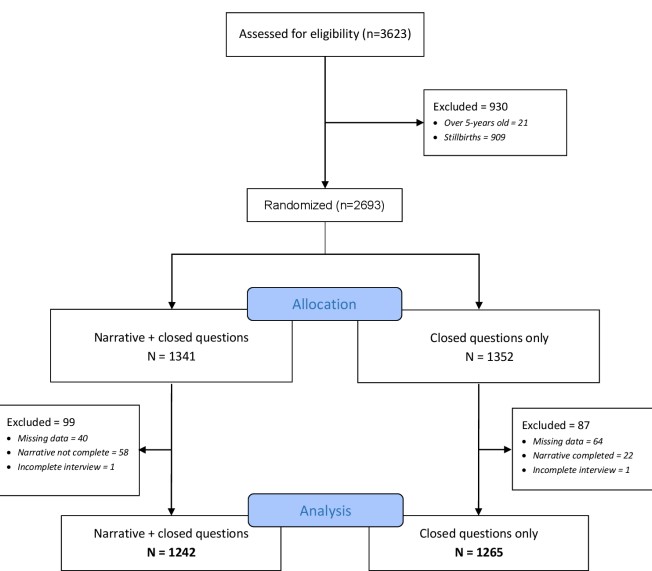

**Figure 1** Verbal autopsy inclusion (Consolidated Standards of Reporting Trials diagram).

discussion was conducted in English, audio-recorded and transcribed verbatim. The data were coded by CK using pen and paper, with predefined deductive themes using a thematic approach. The themes were based on the aim of the study: the interview process and procedures; perceived data quality and emotional stress during VAs. The final interpretation was shared with the MEOs, after triangulation with the quantitative analysis to check that it corresponded with their experiences.

### Patient and public involvement

Prior to community VAs beginning, the overall VacSurv Study protocol was presented to the District Executive Committee and District Health Management team in Mchinji for input and approval. Extensive community engagement was conducted before data collection, and continued throughout the study, through village-level key informant volunteers, area development committees and radio jingles. Community consent from traditional leaders was sought during study introduction.

### RESULTS

A total of 3623 under-5 deaths were reported during the study period, and 2507 were included in the analysis (figure 1). Overall, 50.2% (n=1352) were allocated to no narrative and 49.8% (n=1341) to have an open narrative, with 95% and 94% of VAs conducted per-protocol in each arm (online supplemental appendix 2). Balance in the randomisation was achieved for respondent type, socioeconomic status, child sex and time since the death. However, more open narrative interviews were conducted for neonates (44.5% vs 39.9%, p=0.042) and location of death differed between the two groups (online supplemental appendix 3). Of the deaths, 41.9% were neonates, 52.9% were boys and 31.8% occurred at home (online supplemental appendix 3). Primary respondents were

**Table 1** Description of VA duration

| | No narrative N=1265 | Narrative N=1242 | P value |
|---|---|---|---|
| Total VA duration (min) | | | |
| <10 | 259 (20.5%) | 33 (2.7%) | |
| 11–20 | 663 (52.4%) | 381 (30.7) | |
| 21–30 | 252 (19.9%) | 422 (34.0%) | |
| >30 | 66 (5.2%) | 401 (32.3%) | |
| Missing | 25 (2.0%) | 5 (0.4%) | <0.001 |
| Closed question duration (min)* | | | |
| Min–max | 6–134 | 6–164 | |
| Median (IQR) | 15.0 (9.7) | 19.8 (9.9) | <0.001 |
| Narrative duration (min) | | | |
| Min–max | | 1–113 | |
| Median (IQR) | | 7.0 (5.0) | |

*The duration included pauses in the interview; MEOs were instructed to allow at least 30 min if the respondent needed to pause the interview before attempting to reschedule. Note, only 17 (0.7%) interviews took longer than 60 min.
MEOs, monitoring and evaluation officers; VA, verbal autopsy.

mainly mothers (77.0%, n=1931), followed by grandparents (10.6%, n=266) and fathers (8.0%, n=200). The mean time between death and verbal autopsy was 22.5 weeks (range: 1–52 weeks). We present the quantitative and qualitative results together under the following themes: VA processes, data quality and emotional stress.

### VA processes and procedures
Open narratives took a median of 7 minutes (range: 1–113 minutes) and closed questions took 17.5 minutes (range: 6–164 minutes). Overall, interviews which included narratives took longer to complete, with 32.3% taking longer than 30 min compared with only 5.2% of those without a narrative (p<0.001), with the accompanying closed questions correspondingly taking longer on average to complete (table 1).

From the group discussion, MEOs reported narratives generally taking between 3 and 15 minutes. A key factor in the duration of these was the respondent and whether they were capable and willing to respond. Respondents who were keen to relay their story were reported to do so without prompting, including in interviews randomised to not include a narrative. Conversely, respondents who were hard to engage in interviews with a narrative were also reported.

> My experience has been that after getting consent sometimes a respondent starts to recount before you ask, so you don't interrupt, you just listen. But because your phone has asked you not to take an open history, you don't take notes on that, you just go straight to the questions. (MEO 8)

> And you can see that there were some open histories that were very short, maybe just 2 minutes [general agreement]. You just know that the respondent was

not ready to give you information. It happened like that. (MEO 2)

However, in general the MEOs agreed that the main difference in interviews with and without open narratives was the time taken. Not taken into consideration in the quantitative measures of duration was the time to identify appropriate respondents before an interview could start. This could involve multiple visits to a respondent's household before an appropriate respondent could be located (eg, mother or father), or needing to gain community trust to access the respondent.

> But some other times it may take even 10 minutes because these people know who you want to talk to, but they are trying to shield them because they are not very sure at first what you've come to do. (MEO 1)

When asked what they would recommend as the best VA procedure, there was a consensus that both the open narrative and closed questions were important and should be included: 'The best way is the one which has the open history, that way you have the full explanation' (MEO 7).

### Data quality
Based on InterVA analysis of closed questions, 94.3% of deaths had a cause of death assigned; there was no difference between those with and without an open narrative (94.7% vs 93.9%, p=0.404). Comparing the number of positive responses in the closed questions found no differences with a mean of 22.4, 21.4 and 21.8 'yes' answers for neonates, infant and child VA interviews (table 2). The addition of the open narrative was not associated with respondents expressing a desire to know or suggest a potential cause of death.

**Table 2** Description of respondent emotional stress and VA quality indicators, between interviews with and without open narratives

| | No narrative n (%) | Narrative n (%) | Total n (%) | P value |
|---|---|---|---|---|
| **Respondent displayed visible emotional distress** | | | | |
| No | 1042 (82.4) | 990 (79.7) | 2032 (81.0) | |
| Yes | 223 (17.6) | 252 (20.3) | 475 (19.0) | 0.089 |
| Type of emotional distress displayed during interview* | | | | |
| Crying | 4 (1.8) | 12 (4.8) | 18 (3.4) | |
| Long silence | 59 (26.5) | 68 (27.0) | 127 (26.7) | |
| Other signs of emotional distress | 160 (71.8) | 172 (68.2) | 332 (69.9) | 0.191 |
| Interview paused due to respondent becoming emotionally distressed* | | | | |
| No | 89 (39.9) | 117 (46.4) | 206 (43.4) | |
| Yes—once | 31 (13.9) | 41 (16.3) | 72 (15.2) | |
| Yes—more than once | 103 (46.2) | 94 (37.3) | 197 (41.5) | 0.146 |
| **Respondent expressed desire to know the cause of death** | | | | |
| No | 1235 (97.6) | 1216 (97.9) | 2451 (97.8) | |
| Yes | 30 (2.4) | 26 (2.1) | 56 (2.2) | 0.638 |
| **Respondent suggested potential cause of death** | | | | |
| No | 909 (71.9) | 890 (71.7) | 1799 (71.8) | |
| Yes | 356 (28.1) | 352 (28.3) | 708 (28.2) | 0.912 |
| InterVA able to assign cause of death | | | | |
| Indeterminate | 77 (6.1) | 66 (5.3) | 143 (5.7) | |
| Determinate | 1188 (93.9) | 1176 (94.7) | 2364 (94.3) | 0.404 |
| Number of 'yes' responses to closed questions† | **Mean (SD)** | | | |
| Neonates | 22.6 (5.3) | 22.2 (5.3) | 22.4 (5.3) | 0.297 |
| Infants | 21.5 (6.8) | 21.3 (7.3) | 21.4 (7.0) | 0.658 |
| Child | 22.2 (8.2) | 21.3 (8.1) | 21.8 (8.1) | 0.122 |

*Questions only asked for respondents who had a visible display of emotional distress (n=475); 'other' was not further specified, but informal feedback from MEOs reported examples of distress seen in facial expressions and body language.
†Different numbers of questions are asked for different age groups.
MEOs, monitoring and evaluation officers; VA, verbal autopsy.

There was consensus from the MEOs that data collected were of better quality when they conducted an open narrative. The first reason was that they effectively asked the questions two times, once as the narrative and then a second time in the closed questions, enabling them to cross-check responses. Second, MEOs reported respondents being more comfortable narrating a story than responding to 'yes/no' questions. Finally, they reported the information gained during the narrative helped them navigate through the closed questions and probe respondents for details in a more directed fashion.

I have that feeling that, without the open history, the quality is compromised. Because it's like the recall system, the set-up of the brain of the respondent, is disturbed by questions time and again. Unlike when he or she is free to express everything from her memory, it happens to be good quality data […] I think that open history gives a respondent a feeling that

you are really concerned, because you take a lot of time to listen to him or her. (MEO 8)

While only 28.2% of respondents were recorded as providing a cause of death (table 2), the MEOs noted that caregivers would often give a reason for their child's death—especially if they had sought care. However, they also noted that cause of death was not limited to medical reasons:

In their narrations, they will tell you the cause, 'yes this baby was suffering from malaria, but we think this baby died because they delayed in referring us to a health centre'. Maybe in the most remote areas there was no ambulance, they were told to come to the [town] but the ambulance was not available. They were told to look for their own transport to the [town]. So they will tell you those ones as reasons, not the actual sickness of the baby. (MEO 4)

**Table 3** Logistic regression exploring associations between respondent and child characteristics and emotional distress during VA

**Visible emotional distress due to open narrative**

| Descriptors | | aOR* (95% CI) | P value |
|---|---|---|---|
| Open narrative | No | 1.00 | |
| | Yes | 1.20 (0.98 to 1.47) | 0.084 |
| Respondent | Mother | 1.00 | |
| | Father | 0.72 (0.49 to 1.07) | 0.102 |
| | Grandparent | 0.23 (0.13 to 0.39) | <0.001 |
| | Others | 0.04 (0.01 to 0.28) | 0.001 |
| Child's age | Neonate | 1.00 | |
| | Infant | 1.42 (1.09 to 1.85) | 0.010 |
| | Child under-5 | 1.21 (0.86 to 1.69) | 0.274 |
| Child's sex | Male | 1.00 | |
| | Female | 0.99 (0.80 to 1.22) | 0.920 |
| Location of death | Home | 1.00 | |
| | Health centre | 1.36 (1.04 to 1.77) | 0.023 |
| | MDH | 0.96 (0.72 to 1.27) | 0.753 |
| | En route to hospital | 1.49 (1.00 to 2.22) | 0.049 |
| | Other | 0.38 (0.23 to 0.64) | <0.001 |
| Socioeconomic status by tercile | Tercile 1 (lowest) | 1.00 | |
| | Tercile 2 (middle) | 1.52 (1.17 to 1.97) | 0.002 |
| | Tercile 3 (highest) | 1.49 (1.13 to 1.95) | 0.004 |
| Delay between death and VA (weeks) | | 0.98 (0.98 to 0.99) | 0.002 |

*All variables presented were included in the adjusted analysis.
aOR, adjusted OR; MDH, Mchinji District Hospital; VA, verbal autopsy.

## Emotional stress

In the majority of interviews, respondents did not display visible signs of emotional distress (81%), with similar proportions between those with and without an open narrative (79.7% vs 82.4%, p=0.089). Of those who were recorded as showing signs of emotional distress, 3.4% cried, 26.7% had a long silence and 69.9% exhibited other signs of emotional distress—over half of these interviews needed to be paused once or more (table 2).

Table 3 shows the logistic regression for respondent emotional distress. While having an open narrative was associated with 20% (adjusted OR (aOR): 1.20; 95% CI: 0.98 to 1.47) higher odds of the respondent becoming emotionally distressed during the interview, this was not statistically significant but may be pragmatically relevant. Factors associated with lower odds of becoming emotionally stressed during the VA interview included: non-parental respondents and increased time between the death and interview (2% lower odds for each week passed). Factors associated with increased odds of visible signs of emotional stress include: deaths among infants compared with neonates (aOR: 1.42; 95% CI: 1.09 to 1.85); the death occurring at a health centre (aOR: 1.36; 95% CI: 1.04 to 1.77) or en route to hospital (aOR: 1.49; 95% CI: 1.00 to 2.22); and being in the middle (aOR: 1.52; 95% CI: 1.17 to 1.97) or highest wealth tercile (aOR: 1.49; 95% CI: 1.13 to 1.95).

Respondents' emotional stress was not directly raised by the MEOs during the discussion; however, they noted a key challenge in conducting the VAs as being unable to help respondents or feeling hopeless when respondents related their stories. They raised specific examples around HIV-positive respondents seeking advice or requests for referrals of malnourished children to non-governmental organisation programmes.

> A challenge, in a nut shell, was not being able assist where questions were raised. You have raised questions to them. In the end they raise questions to you, that need action, for you to not be able to do anything. That was a big challenge and a let-down. (MEO 4)

The MEOs indicated that the VA process is similarly distressing for the interviewer, with many of the MEOs also having families and young children who can relate to the narrative.

> The verbal autopsies are not easy to be carried as they involve or concern somebody who has lost life,

so it's always emotional between the interviewer and the interviewee. (MEO 2)

## DISCUSSION

Using a mixed-methods analysis of VA process data among children under-5 in Malawi, we explored the role of open narratives on the interview process, data quality and respondents' emotional stress. As expected, free-text narratives increased the duration of the VA interview but did not impact on the ability of a Bayesian algorithm to assign a cause of death—the proxy we used for data quality. The interviewers considered the open narrative useful in building rapport with respondents, agreeing with previously reported experiences,[11 26] and believed it subsequently improved the VA data. However, respondents with an open narrative displayed emotional distress more frequently when compared with those without, even if relatively uncommon. While it was outside the scope of this study, further work is warranted from the respondent perspective; in particular, whether they value the space to narrate their stories and how this balances with the emotional burden.

Although previous studies have observed VA-induced emotional stress among respondents,[16–18 27] exploring characteristics of both the respondent and deceased showed interesting associations with emotional distress. First, infant death was associated with increased emotional stress during VA compared with neonatal deaths. Grief is influenced by cultural constructs, and here cultural perceptions of child 'maturity' may be important. Studies from Tanzania and Ghana both point to norms around concealing mourning for young infants, in particular pregnancy loss.[28 29] Respondents from the higher socioeconomic group had lower odds of observed emotional stress. Under-5 deaths are more frequent among lower socioeconomic households[30]; with an under-5 mortality of 52/1000 live births in the highest wealth group, compared with 69/1000 in middle and low wealth groups in Malawi.[31] The 'unexpectedness' of deaths among children has been associated with increased parental grief previously[32] ; and a study from South Africa reported that pressures of poverty can overshadow the grieving process.[33] More understanding on how local contexts and mourning processes can affect the VA procedure would be valuable.

We observed that deaths occurring at health centres or en route to hospital were associated with increased emotional stress. This may reflect respondents' perception or experience of poor quality of care, resulting in frustrations at system failures and delays in referrals and receiving care. This was echoed by the MEOs, and prior data from this setting,[12] who described respondents attributing deaths to non-medical causes. Deficiencies in Malawian healthcare facilities' ability to deliver quality maternal, newborn and child care have been reported,[34] and modelled estimates suggest that poor quality maternal and newborn care results in considerable preventable mortality.[35] Caregiver frustration with healthcare provision and challenges in reaching referral facilities are therefore understandable.

Although the MEOs perceived better rapport and improved data quality from VAs with open narratives, we did not observe any differences in the number of 'yes' responses and the subsequent proportion of VAs with an assigned cause of death. Earlier findings from Malawi showed limited advantage in including unstructured open narratives to assign cause of death.[12] While in this case it is hard to know whether individual answers would have been different, comparing cause of death distributions between those with and without narratives found no clear differences (data not shown), suggesting this was not the case. The added diagnostic value of free-text narratives has been examined previously[19] and found that the addition of the narrative did not explain discrepancies in diagnoses between physician and InterVA analyses. This could be due to narratives capturing indicators which are already included in closed questions. A key principle in research ethics is to avoid intrusions[36]; therefore, if narrative data are not intended for analysis and do not appear to have any influence on data quality, documenting these data may pose an unnecessary burden.

The main limitation of our study was our reliance on interviewer-observed signs of respondent emotion. The MEO self-completed post-VA questionnaire may have suffered from the subjective nature of expression and interpretation of emotion, and cultural norms of private bereavement. Grief after the death of an infant has also been described as a non-linear process and influenced by gender.[37] Including questions on both respondent-reported and interviewer self-reported emotional stress would have provided richer information. The MEOs also reported being upset by the VA interviews. A study from Mexico has developed a containment strategy to support the emotional health of data collectors conducting VAs, and going forward this should be considered.[38]

It has been reported that women can face stigma and blame in child deaths.[39] The power imbalance and gendered interviewer–interviewee dynamics present in this study may therefore have influenced mothers' emotional stress and willingness to freely discuss their child's death. These dynamics may also be present when multiple respondents were interviewed together, for example, a husband and wife. While we allowed the main respondent to decide who else was present for the VA, women may not have been empowered to exclude others from the process. The project conducted extensive community sensitisation through working with village leaders and key informants to gain respondent trust. However, the MEOs being local residents may also have inadvertently affected this, as Haws *et al* found interviewers from outside the community with good cultural insights may be more trusted.[29]

It is likely that undocumented protocol violations occurred, as MEOs reported respondents being

unwilling or unable to fully engage in the open narrative, and conversely narrating the story of their child's death without prompt. This is not unlike the reluctance observed in VA respondents in rural Ghana who occasionally denied interviews due to grief.[15] While we planned a per-protocol analysis, we were unable to fully adjust for these violations in the quantitative analysis, and our results may therefore more closely reflect intention-to-treat. Finally, the group discussion with the MEOs was led by the technical advisor, possibly leading to social-desirability bias limiting their willingness to highlight concerns or deviations from the protocol.

## CONCLUSION

Evidence from this large-scale evaluation suggests that open narratives do not affect the ability of an automated algorithm to assign a cause of death, but play a valued role in establishing interviewer–interviewee rapport. From the interviewer perspective, good rapport as a result of conducting an open narrative at the start of the VA may outweigh the additional time burden and the slight increase in emotional stress among respondents. Any undue burden associated with having an open narrative would be further justified if the quality and utility of information can be guaranteed. We would therefore recommend guidance from leading bodies, such as the WHO VA Reference Group, for a more standardised approach to record and analyse free-text narratives. This is with a view to reducing bias introduced by those involved during the VA process, but should also take into account respondent perspectives and preferences. We would also support longer waiting periods between death and time of interview, so long as accurate recall is not negatively impacted, and the inclusion of wider non-parental family members to reduce the emotional burdens associated with the sensitive nature of discussing death.

**Correction notice** This article was previously published with wrong licence. The correct licence for the paper is CC-BY.

**Acknowledgements** We would like to thank the communities and families participating in the study, the key informants for volunteering their time and the Traditional Authorities of Mchinji district and Chilumba, Karonga district for their support. We are very grateful for the hard work of our field and data staff. We thank the other VacSurv Consortium members: Neil French and Nigel Cunliffe (University of Liverpool); Rob Heydermann and Anthony Costello (University College London); Amelia Crampin (University of Glasgow); Osamu Nakagomi (University of Nagasaki); Jacqueline E Tate and Umesh D Parashar (Centers for Disease Control and Prevention, Atlanta, Georgia, USA).

**Contributors** This study was conceived by PL, EF and CK, based on data from a larger research question conceived by NB-Z, BJB, JB, EF and CK. The data were collected under the oversight of TP, CM and MB. The quantitative data were analysed by PL and the qualitative data were coded and analysed by CK. The manuscript was written by PL, with significant input from CK. All authors read, commented and approved the manuscript.

**Funding** This VacSurv Project was funded by a Wellcome Trust Programme Grant (WT091909/B/10/Z) to Neil French, Nigel Cunliffe, Rob Heyderman; an investigator-initiated grant by GlaxoSmithKline Biologicals to Nigel Cunliffe, NB-Z, Neil French; a Wellcome Trust Strategic Award to Anthony Costello (number: 085417ma/Z/08/Z); and MLW core grant strategic award to Rob Heyderman.

**Disclaimer** The findings and conclusions in this report are those of the authors and do not necessarily represent the official position of the US Centers for Disease Control and Prevention.

**Competing interests** None declared.

**Patient consent for publication** Not required.

**Ethics approval** The study was approved by the National Health Sciences Research Ethics Committee in Malawi (#837), London School of Hygiene and Tropical Medicine, UK (#6047) and Centers for Disease Control and Prevention, USA (#6268). Verbal informed consent was sought from respondents prior to starting the verbal autopsy interviews. Consent was documented in the electronic data capture form. Written informed consent was obtained from the MEOs prior to the group discussion.

**Provenance and peer review** Not commissioned; externally peer reviewed.

**Data availability statement** Data are available upon reasonable request. Fully anonymised quantitative datasets generated and analysed for the purpose of this study are available from the corresponding author, CK (c.king@ucl.ac.uk) on reasonable request for research purposes only, following approval from the National Health Sciences Research Ethics Committee in Malawi, and study principal investigators.

**ORCID iD**
Carina King http://orcid.org/0000-0002-6885-6716

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
