## [Reviewer comments · BMJ Paediatrics Open]

ARTICLE DETAILS

TITLE (PROVISIONAL)	The added value of an open narrative in verbal autopsies – a mixed-methods evaluation from Malawi
AUTHORS	Loh, Patricia Fottrell, Edward Beard, James Bar-Zeev, Naor Phiri, Tambosi Banda, Masford Makwenda, Charles Bird, Jon King, Carina

VERSION 1 – REVIEW

REVIEWER	Reviewer name: Dr. Conrad Kabali Institution and Country: 2264 Spence Lane, Burlington, Ontario L7L6L3, Canada Competing interests:None
REVIEW RETURNED	28-Nov-2020

GENERAL COMMENTS	The authors evaluated the value of adding open narrative questions in verbal autopsy. Overall, the manuscript is well written. I have a couple of comments. Page 5, line 36: The objective of "blinding the randomization procedure" is unclear. What kind of bias were the authors trying to prevent? This needs to be clarified. Page 6, line 15: The comparison is between the means or proportions. Tests such as student's t or chi-square are added after this comparison has been made (they are not informative enough on their own) Page 6, line 21: Is it likelihood or posterior probability? Page 6, line 28: How often were interviews excluded? if that happened too frequently, randomization will be broken. I will advise the authors to provide a table that compares the distribution of potential confounders between the per-protocol population and non-compliers. Page 6, line 33: Please add potential confounding variables into table 1. The title will need to be modified reflect the changes. Page 9, line 14: The percentage of people who displayed visible signs of emotions was quite high (~19%), meaning that the odds ratio may have overestimated the risk ratio. If the authors wish to make a comparison of risks/probabilities, I will advise that they directly compute the risk ratios. Otherwise, the odds ratio in this case should strictly be interpreted as the comparison of odds
--

REVIEWER	Reviewer name: Omrana Pasha Razzak Institution and Country: Penn State, United Kingdom of Great Britain and Northern Ireland Competing interests: None
-----------------	--

REVIEW RETURNED	18-Nov-2020
-------------

GENERAL COMMENTS	I congratulate the authors for this well-written and interesting article. Below find some comments which could improve readers' understanding of the research presented.  1. Consider emphasizing that the narrative was not used for assignment of cause of death early in the write-up. In fact, it may even be important to clarify this in the abstract. 2. Please provide details about sample size determination and the a priori calculations of power of your study 3. You have chosen to use per protocol analysis. While this is perfectly acceptable, it may be useful to discuss why that choice was made and what difference intent-to-treat analysis would have made. 4. The authors may consider replacing the phrase "respondent emotion" with "emotional stress amongst respondents" or "emotional distress" or "distress". I reviewed the literature for this concept and found these to be the most commonly used terms. It also appears to me that these terms are also more descriptive than "emotions" 5. Greater detail on the methods for the qualitative arm of the study could make it clear 6. how these data were analyzed. 6. In the conclusion, (page 12; line 18) you note that "We would also support longer waiting periods between death and time of interview, so long as accurate recall is not negatively impacted, and the inclusion of wider nonparental family members to reduce the emotional burdens associated with the sensitive nature of discussing death." It is not clear to me what part of your data you are basing these conclusions on and would recommend greater clarification on this. 7. While the authors appear to have followed all necessary ethics procedures, I would advise one change, which I believe is due to the way it is written, rather than an ethical lapse. Correcting that would make it clear that there was no ethical impropriety. In line 37/38 on page 6 the authors write: "The respondent was blinded to the randomisation procedure, but MEOs were unblinded to the purpose of randomisation." It is my assumption that the authors informed respondent's about the use of random assignment to obtain informed consent from study participants. The study participants, while aware of the random assignment used in the study were not aware of their own, individual assignment. The above line needs to be edited for language to clarify this. I hope that my comments are helpful to the authors.
--

REVIEWER	Reviewer name: Dr. Anita Chary Institution and Country: Not applicable Competing interests: None
-----------------	---

REVIEW RETURNED	29-Nov-2020
-------------

GENERAL COMMENTS	Reviewing this article as a medical anthropologist and physician, my major concern relates to the idea that emotions can be rated through visualization (as some emotional cues are not visual) and
--

	the article's basis, in part, on subjective interpretation of another person's behaviors during an interview. To the authors' credit, they allude to the cultural nature of emotional expression. I think that is a significant methodological limitation. There is presumably some difference in literacy, SES between the interviewers and respondents, which would affect perception of emotions. The respondents had different ages/gender identities/relationships to the deceased which would have presumably affected cultural norms around emotional display. This wasn't the point of the study but there are different emotional reactions culturally to death of a male vs female child (with the former tending to be mourned to a different extent than the latter in some societies). The time range of interview occurring 1-52 weeks after a death means reaching respondents at different stages of coping. These issues should be addressed in the introduction, methods, and conclusion/limitations section in a fuller way. There is a robust social science literature about gender and family roles in Malawi which can be cited. There is also psychological and anthropological literature about how cultural values affect the expression and interpretation of emotion which could be cited and discussed briefly. I would request a bit more detail about the SES of the interviewers and how this compares to the population of respondents. I would also request within the table and a few words within the manuscript more detail about the category of "other" emotions--which represent >60% of what was documented. What could be further clarified is the benefit of rapport during the interview. Perceived increase in rapport makes the interview more comfortable for the interviewer--but this study doesn't indicate whether this is perceived to add value by the respondents. Presumably some of them would prefer to keep the interaction to minimal timing as it's uncomfortable to some degree regardless of rapport. Why exactly would interviewer comfort/perception of added value be enough to justify continuing the verbal autopsy? Another recommendation is a stronger statement about whether the authors think maintaining the VA is actually justifiable/recommendable. There are some areas where the authors seem to support its inclusion and other areas where the text reads as hedging about continued inclusion.
--	---

REVIEWER	Reviewer name: Dr. Dolores Ramirez-Villalobos Institution and Country: Instituto Nacional de Salud Publica, Centro de Investigaciones en Sistemas de Salud, Mexico Competing interests: None
REVIEW RETURNED	07-Dec-2020

GENERAL COMMENTS	Despite the fact that the surveyors have been applying Av for 5 years, they were given training on how to perform verbal autopsies on family members or the mother who has lost a child in a crisis. How did they take care of these aspects? Is training mentioned to the interviewers? How was the training to control the crises to the interviewers and the relatives? There is literature that mentions the importance of care when applying Avs: How to deal with the suffering: Utility of an emotional containment strategy to collect data for verbal autopsies in Mexico)
--

(ISSN: 0748-1187 (Print) 1091-7683 (Online) Journal homepage: <https://www.tandfonline.com/loi/udst20>

What was the grieving time given to the family member?

There is no mention of what dates they took care not to cause more pain to the informants

In the methods part it is not specified how they define how the child's primary caregiver was identified, how they cared for and evaluated this. In the Background they mention field workers trained to identify the primary caregiver, and mention the quote (3) that has nothing to do with what is in this paragraph. This should be described in the methodology.

They make mention of adverse effects of induced distress when applying the Av and how this reflects on the quality of data as they controlled for this?

They mention as part of the objective the effects on interview procedures, data quality, and respondent emotion. Their results are based on presenting the differences in 3the type of VA applied

What tools did you implement for ethical care about the emotions of the interviewer and the respondent?

Despite having ethical consent, applying verbal autopsies to family members or mothers of minors is very painful.

How many Avs per interviewer applied, as the Avs were assigned, contemplated the empathy of the interviewers to apply the Av

Did they check the information against the death certificate? What was the standard gold for the cause of death? There is the Tariff Method, which has been evaluated by the Population Health Metrics Research Consortium (PHMRC) gold standard verbal autopsy validation study

How was the assignment made for each group? They mention randomization to signify what type of VA to apply, but how was the assignment made by the respondents?

How many men, how many women of what age, level of education, were health personnel as illness or health promoters)

Because they did not apply the two AV methods to the newborn group.

Since the mother was the main informant, what actions they took in the mother's moments of crisis.

You mention 22 weeks between the event of death and the application of the VA, how do you consider the memory bias to be?

What happened if upon arrival at the home the mother reported another death event in the past month? What actions did they take?

It has been documented that a double event of death can occur in the home.

How do you handle twins?

It is interesting to note the wide range of the application of open narrative up to 164 minutes (what happened? in cases of narratives longer than 2 hours? They mention a value of P in the time differences, but more than knowing that there is a difference, why were those times so long?

	It is not clear how they handled this: in the results the respondents who were difficult to interview with a narrative were also reported. What did they do when the informant was not prepared to give the information, even if the mourning was short? what happened? it is a crucial point for the care of the emotions and feelings of the informant even more they mention that more than 70% was the mother What do they refer to with appropriate respondents before the interview begins? If their objective was to compare the effects and emotions of applying a closed-ended and open-ended SD, they should have taken more care in selecting the interviewers and the training they provided, rather than the technical training of the instrument. The differences in the timing of the two methods are clear, but what was the objective of comparing which is better, when the VA instrument is already standardized by WHO and validated by research conducted by the Population Health Metrics Research Consortium (PHMRC) gold standard verbal autopsy validation study (Serina, P., Riley, I., Stewart, A. et al. A shortened verbal autopsy instrument for use in routine mortality surveillance systems. BMC Med 13, 302 (2015). https://doi.org/10.1186/s12916-015-0528-8; Serina, P., Riley, I., Stewart, A. et al. Improving performance of the Tariff Method for assigning causes of death to verbal autopsies. BMC Med 13, 291 (2015). https://doi.org/10.1186/s12916-015-0527-9
--	---

VERSION 1 – AUTHOR RESPONSE

Please note, all page and line numbers refer to the marked up version of the manuscript.

Reviewer: 1

The authors evaluated the value of adding open narrative questions in verbal autopsy. Overall, the manuscript is well written. I have a couple of comments.

We thank the reviewer for their kind review and constructive feedback.

1. Page 5, line 36: The objective of "blinding the randomization procedure" is unclear. What kind of bias were the authors trying to prevent? This needs to be clarified.

We have now included examples of biases we hoped to minimise through blinding the respondents of the randomisation (pg. 5, line 134).

2. Page 6, line 15: The comparison is between the means or proportions. Tests such as student's t or chi-square are added after this comparison has been made (they are not informative enough on their own)

We have re-phrased this in the methods (pg. 6, line 163).

3. Page 6, line 21: Is it likelihood or posterior probability?

Thank you for noting this – InterVA assigns posterior probability and we have amended the text to reflect this.

4. Page 6, line 28: How often were interviews excluded? if that happened too frequently, randomization will be broken. I will advise the authors to provide a table that compares the distribution of potential confounders between the per-protocol population and non-compliers.

We have now included an additional supplementary table to show differences between those who complied and those who did not (WebAppendix 2). Overall only 5% and 6% of interviews were excluded in each arm, respectively.

5. Page 6, line 33: Please add potential confounding variables into table 1. The title will need to be modified reflect the changes.

Table 1 presents a summary of the duration of interviews, comparing those with and without a narrative. We are not sure adding confounders for this would be informative. More data on the possible confounders are presented in the table in WebAppendix 3. Please let us know if we have misunderstood this comment and further edits are required.

6. Page 9, line 14: The percentage of people who displayed visible signs of emotions was quite high (~19%), meaning that the odds ratio may have overestimated the risk ratio. If the authors wish to make a comparison of risks/probabilities, I will advise that they directly compute the risk ratios. Otherwise, the odds ratio in this case should strictly be interpreted as the comparison of odds.

We have re-checked the manuscript to ensure that odds ratios are consistently reported in terms of odds and not risks.

Reviewer: 2

Reviewing this article as a medical anthropologist and physician, my major concern relates to the idea that emotions can be rated through visualization (as some emotional cues are not visual) and the article's basis, in part, on subjective interpretation of another person's behaviours during an interview. To the authors' credit, they allude to the cultural nature of emotional expression. I think that is a significant methodological limitation. There is presumably some difference in literacy, SES between the interviewers and respondents, which would affect perception of emotions. The respondents had different ages/gender identities/relationships to the deceased which would have presumably affected cultural norms around emotional display. This wasn't the point of the study but there are different emotional reactions culturally to death of a male vs female child (with the former tending to be mourned to a different extent than the latter in some societies). The time range of interview occurring 1-52 weeks after a death means reaching respondents at different stages of coping. These issues should be addressed in the introduction, methods, and conclusion/limitations section in a fuller way.

We would like to thank the reviewer for bringing a different perspective, and highlighting several areas that we could address more fully. We have tried to expand on specific recommendations given and add a more reflexive tone in the discussion.

1. There is a robust social science literature about gender and family roles in Malawi which can be cited.

We have incorporated this into the discussion and limitations in more detail, while trying to keep the discussion (and overall paper!) concise, as we agree, gendered constructs are important in both how grief is expressed and in the dynamic between interviewer and interviewee (pg 11).

2. There is also psychological and anthropological literature about how cultural values affect the expression and interpretation of emotion which could be cited and discussed briefly.

We have now incorporated more discussion on the cultural constructs of grief in the discussion – given this is a much wider topic area, we appreciate that we have only touched on it briefly (pg 10, line 327).

3. I would request a bit more detail about the SES of the interviewers and how this compares to the population of respondents.

The MEOs who conducted the interviews were all Malawian, male and resided within the study area. They all had a high school diploma and several had completed courses in community mobilisation and social work. Therefore, in many of the interviews, there would be key demographic differences between the respondent and interviewer (gender, education). However, interviewers lived in similar contexts and many had young children, which would offer shared cultural understanding. The MEOs had all worked within these communities for over 5 years before starting these VAs, and the local NGO organisation was well known and trusted in communities. We have added more detail on this in the Methods and Discussion (pg 5 and pg 11).

4. I would also request within the table and a few words within the manuscript more detail about the category of "other" emotions--which represent >60% of what was documented.

We have added a statement to the footnote of Table 2.

5. What could be further clarified is the benefit of rapport during the interview. Perceived increase in rapport makes the interview more comfortable for the interviewer--but this study doesn't indicate whether this is perceived to add value by the respondents. Presumably some of them would prefer to keep the interaction to minimal timing as it's uncomfortable to some degree regardless of rapport. Why exactly would interviewer comfort/perception of added value be enough to justify continuing the verbal autopsy?

In terms of the value of rapport, we viewed this as an important factor in data quality, allowing interviewers to probe more freely and use closed questions are a way to verify the information from narratives. We have now added an additional sentence to the results on this (pg. 8, line 263). In the interest of word counts, we did include the full quote, but can add the following to the one from MEO 8 in the text: "Some of the questions were redirecting the person to say yes or no, whilst the open history, one simple question will give you almost everything, you will just need to probe using the other questions after the open history."

However, it is a very valid point about added-value to respondents, and one which we did not evaluate. From informal field reports during the project, it was noted that some respondents welcomed the process of a narrative, as it showed someone was interested in their experiences and their child. This was also captured in the group discussion by respondents starting to tell their story before being prompted, and we have a quote included on this. We have added a statement to expand on this in the discussion (pg. 10, line 320) – but acknowledge that further work on understanding respondent preferences is very much warranted.

6. Another recommendation is a stronger statement about whether the authors think maintaining the VA is actually justifiable/recommendable. There are some areas where the authors seem to support its inclusion and other areas where the text reads as hedging about continued inclusion.

Based on the reviewer's comments, we have edited the conclusion to more clearly reflect our recommendation that while narratives do not appear to affect automated assignment of cause of death, that the building of rapport may still be important for data quality. We

have also added emphasis that any associated burdens elicited by the narrative must be justified by ensuring quality and utility of information from the narrative which can be achieved through standardised recommendations from leading bodies such as the WHO VA Reference Group.

Reviewer: 3

Despite the fact that the surveyors have been applying VA for 5 years, they were given training on how to perform verbal autopsies on family members or the mother who has lost a child in a crisis. How did they take care of these aspects? Is training mentioned to the interviewers? How was the training to control the crises to the interviewers and the relatives? There is literature that mentions the importance of care when applying VAs (How to deal with the suffering: Utility of an emotional containment strategy to collect data for verbal autopsies in Mexico; <https://www.tandfonline.com/loi/udst20>)

The Standard Operating Procedures for the VAs, including how to manage respondent distress and personal emotional distress, were developed together in discussion during the training. The interviewers (many of whom had young children themselves), drew on their prior experience to develop pragmatic and locally appropriate strategies. This covered aspects such as offering condolences, how long to pause an interview before stopping, how to find the most appropriate respondents. We have added a statement on this in the methods (pg 5, line 149).

However, in terms of support for the interviewers, we did not have a formal procedure set up. Informal processes were present, and the technical advisor and senior M&E manager had frequent feedback discussions with the interviewers. Given our results, we agree, this is an important consideration, and we have now added a sentence on this in the discussion – and included this reference (pg 11, line 373).

What was the grieving time given to the family member?

The average time between death and interview was 22 weeks, and we aimed as much as possible to leave a minimum of 2-weeks between death and interview, in line with recommendations. In practice the time between death and VA was determined by the data management processes for the vital events surveillance system in place.

There is no mention of what dates they took care not to cause more pain to the informants.

We were not clear on the information the reviewer would like us to add or clarify.

In the methods part it is not specified how they define how the child's primary caregiver was identified, how they cared for and evaluated this. In the Background they mention field workers trained to identify the primary caregiver, and mention the quote (3) that has nothing to do with what is in this paragraph. This should be described in the methodology.

Reference 3 was acknowledged as part of the sentence as it had described the two stages of VA – the first being trained fieldworkers interviewing final caregivers, usually relatives. We have removed this, as we have referenced WHO's VA instrument (reference 8). For clarity, in this context, if the mother is alive and present in the household they were considered the primary caregiver, along with the father. Grandparents and older siblings also provide a caring role in this context, and were therefore considered eligible respondents.

They make mention of adverse effects of induced distress when applying the VA and how this reflects on the quality of data as they controlled for this?

We hypothesised that the narrative may affect respondents' subsequent responses to the ensuing closed questionnaire which had the options of "Yes", "No" or "Don't know", and would affect the way the interviewers asked subsequent questions. We therefore used the number of "Yes" answers, and subsequently InterVA's ability to assign probable cause of death based on the affirmative answers as a proxmeasure of data quality.

They mention as part of the objective the effects on interview procedures, data quality, and respondent emotion. Their results are based on presenting the differences in the type of VA applied. What tools did you implement for ethical care about the emotions of the interviewer and the respondent?

As presented in our response above, we developed locally appropriate SOPs collectively. MEOs were encouraged to raise challenges they faced during routine monthly team meetings, including ways to improve working.

Despite having ethical consent, applying verbal autopsies to family members or mothers of minors is very painful. How many VAs per interviewer applied, as the VAs were assigned, contemplated the empathy of the interviewers to apply the VA.

VAs were assigned to the different interviewers according to geographical area – each interviewer was responsible for a specific catchment area. This was done to ensure they were familiar and known to their communities, village leaders and key informants. As the catchment areas were of roughly equal population size, the numbers of VAs conducted by each MEO was relatively similar.

Did they check the information against the death certificate? What was the standard gold for the cause of death? There is the Tariff Method, which has been evaluated by the Population Health Metrics Research Consortium (PHMRC) gold standard verbal autopsy validation study

In this context there was no formal death registration or process of death certification, hence the need to conduct the verbal autopsies. It was outside the scope of this study to attempt to determine the accuracy or validity of the assigned cause of death, and others have looked at this question previously – we have clarified this in the methods (pg 6, line 174).

How was the assignment made for each group? They mention randomization to signify what type of VA to apply, but how was the assignment made by the respondents?

Randomisation was programmed into the electronic data capture form at the time of commencement of the VA. Respondents were randomised immediately before the VA commenced, and following consent, to receive either WHO 2012 closed questions only or open narrative followed by the closed questions.

How many men, how many women of what age, level of education, were health personnel as illness or health promoters).

We have added more detail on who the interviewers were in the methods (pg. 5, line 142).

Because they did not apply the two VA methods to the newborn group.

To clarify, we only excluded stillbirths from this assessment, but deaths amongst newborns were included in the analysis. This is described in the methods (pg 6, line 173).

Since the mother was the main informant, what actions they took in the mother's moments of crisis.

We developed Standard Operating Procedures for the interviewers to follow in order to ensure interviews were conducted with informed consent by caregivers, and that they

were professional but supportive in cases where respondents became distressed. This included informing caregivers that they could end the interview at any point, and allowing respondents to pause and take a break, or returning at a better time. However, one of the challenges raised by the MEOs was that they couldn't provide more concrete actions which caregivers asked for, which we have highlighted in the manuscript.

You mention 22 weeks between the event of death and the application of the VA, how do you consider the memory bias to be?

We felt it outside the scope of this paper to explore the role of memory bias, but the mean duration between death and interview for the indeterminant deaths was 16 weeks. This suggests that memory bias may not have been a major issue in recalling details of signs and symptoms.

What happened if upon arrival at the home the mother reported another death event in the past month? What actions did they take?

We are not aware of any instances where this occurred – all deaths were reported to the central office by village level key informants, which would then trigger a VA to be conducted. More details of the vital events surveillance have been published here: Bar-Zeev et al., 2015.

It has been documented that a double event of death can occur in the home. How do you handle twins?

We had a few cases where we had reports of deaths in twins. In these cases, two different VA records were documented. If the death had occurred on the same day (e.g. peri-natal deaths) and were reported together, then we could conduct the interview for both deaths simultaneously to minimise the number of questions asked.

It is interesting to note the wide range of the application of open narrative up to 164 minutes (what happened? in cases of narratives longer than 2 hours? They mention a value of P in the time differences, but more than knowing that there is a difference, why were those times so long?

There were only 17 VAs (0.7%) which were longer than 1 hour. We did not keep a formal record of why interviews were longer than others, but MEOs were instructed to allow 30 minutes for the respondent to resume an interview, before attempting to reschedule or stop the process. Therefore, having some interviews take 2.5 hours should not be unexpected. We have added a short note to the Table 1.

It is not clear how they handled this: in the results the respondents who were difficult to interview with a narrative were also reported. What did they do when the informant was not prepared to give the information, even if the mourning was short? what happened? it is a crucial point for the care of the emotions and feelings of the informant even more they mention that more than 70% was the mother

In cases where respondents were quiet, or not sure of answers, the MEOs were trained to asking probing questions to get more information and ask if the respondent wanted anyone else there to help them. But this is a careful balance between probing and be respectful to respondents, and the MEOs used their long experience in conducting VAs and data collection to share good practice with each other. These were raised in the group discussion, but we didn't include any of the quotes in the text due to space.

What do they refer to with appropriate respondents before the interview begins?

An appropriate respondent was identified for the interview as recommended by WHO's standardised VA instrument (2012 version at time of study) i.e: "primary caregiver (usually

a family member) who was with the deceased in the period leading to death or a witness to a sudden death or accident.” We have referenced this guidance.

If their objective was to compare the effects and emotions of applying a closed-ended and open-ended SD, they should have taken more care in selecting the interviewers and the training they provided, rather than the technical training of the instrument. The differences in the timing of the two methods are clear, but what was the objective of comparing which is better, when the VA instrument is already standardized by WHO and validated by research conducted by the Population Health Metrics Research Consortium (PHMRC) gold standard verbal autopsy validation study (Serina, P., Riley, I., Stewart, A. et al. A shortened verbal autopsy instrument for use in routine mortality surveillance systems. BMC Med 13, 302 (2015).

<https://doi.org/10.1186/s12916-015-0528-8>; Serina, P., Riley, I., Stewart, A. et al. Improving performance of the Tariff Method for assigning causes of death to verbal autopsies. BMC Med 13, 291 (2015). <https://doi.org/10.1186/s12916-015-0527-9>

Thank you for highlighting this. The WHO standard VA tool does include an open ended section but its use is debated and is not used universally, and there is a lack of empirical evidence of its value – hence the purpose of our study. As we move towards automated methods and scale-up of VA on a routine basis it is important to make the process as efficient and simple as possible. We were not aiming to comment on whether the information within a narrative can increase the ability to assign cause of death – but rather, whether the process of the narrative effects the quality of data and VA process. We have clarified this in the methods.

Additionally, the interviewers used in the study were selected given their prior experience of conducting VAs, professional work record and their familiarity with communities. As a piece of operation research testing the effect of open-ended parts of VA it would not make sense to select and train interviewers differently. We hope the additions to the methods make this clearer.

Reviewer: 4

I congratulate the authors for this well-written and interesting article. Below find some comments which could improve readers' understanding of the research presented.

We thank the reviewer for their review and constructive recommendations!

1. Consider emphasizing that the narrative was not used for assignment of cause of death early in the write-up. In fact, it may even be important to clarify this in the abstract.

Given the word limit, we struggled to include this explicitly in the abstract, but have added “automated algorithm”. The use of InterVA-4 to assign cause of death for this study is stated in the Methods section (pg 6, line 168) but we agree that emphasising that the narrative was not used for cause of death assignment has been made explicit in the Methods (pg. 6, line 174)

2. Please provide details about sample size determination and the a priori calculations of power of your study

This research question was opportunistically embedded within a larger research study (Bar-Zeev et al., 2015), with the sample size estimate based on determining a reduction in

child mortality associated with vaccine introduction. The sample size was therefore not based on an a priori calculation to answer this research question.

3. You have chosen to use per protocol analysis. While this is perfectly acceptable, it may be useful to discuss why that choice was made and what difference intent-to-treat analysis would have made.

This is a good point, and we have added a statement to explain this in the Methods (pg. 6, line 177). Our main reasoning was that we wanted to explore the mechanism of the narrative on emotion and data quality, rather than the policy of recommending a narrative be conducted or not. In reality, a small proportion of the interviews were not conducted per protocol (5% and 6% in each arm), and therefore the intention-to-treat and per-protocol results were unlikely to differ greatly. We have added a Supplementary Table describing the excluded interviews (Web-Appendix 2).

4. The authors may consider replacing the phrase "respondent emotion" with "emotional stress amongst respondents" or "emotional distress" or "distress". I reviewed the literature for this concept and found these to be the most commonly used terms. It also appears to me that these terms are also more descriptive than "emotions"

Thank you for highlighting this and for the effort in reviewing the literature for this particular concept. We have made the change to emotional stress throughout.

5. Greater detail on the methods for the qualitative arm of the study could make it clear 6. How these data were analysed.

We have added more detail on this to the Methods (pg. 7, line 192).

6. In the conclusion, (page 12; line 18) you note that "We would also support longer waiting periods between death and time of interview, so long as accurate recall is not negatively impacted, and the inclusion of wider nonparental family members to reduce the emotional burdens associated with the sensitive nature of discussing death." It is not clear to me what part of your data you are basing these conclusions on and would recommend greater clarification on this.

This is based on the logistic regression analysis which showed that non-parental respondents and increased time between death and interview (2% lower odds for each week passed) were associated with lower odds of emotional stress amongst respondents (Table 3).

7. While the authors appear to have followed all necessary ethics procedures, I would advise one change, which I believe is due to the way it is written, rather than an ethical lapse. Correcting that would make it clear that there was no ethical impropriety. In line 37/38 on page 6 the authors write:

"The respondent was blinded to the randomisation procedure, but MEOs were unblinded to the purpose of randomisation."

It is my assumption that the authors informed respondent's about the use of random assignment to obtain informed consent from study participants. The study participants, while aware of the random assignment used in the study were not aware of their own, individual assignment. The above line needs to be edited for language to clarify this.

Thank you for raising this point on consent. To clarify, we did not inform respondents of the randomisation process. Randomising participants to within questionnaire variations and questionnaire methodologies is common, and often done without the respondent's knowledge due to the nature of the research question being investigated. In our case, we had two main justifications, which we felt outweighed the potential bias which informing participants of this embedded question might introduce. Firstly, both approaches to the VA are considered standard. Given there is genuine equipoise around the question of the how a narrative effects the interview process, is not clear if either would entail benefits or

harms over the other. Secondly, the information about what taking part in the VA involves is the same for both types of interview, i.e. that we would be asking them about the child's death and an indication of the time needed for the interview. We did not force participants to tell their narrative if they refused, and similarly did not stop people from narrating their story if they wanted to – we have amended the text slightly to highlight that both interviews are standard.

Editorial Comments

This paper generated a lot of interest and some diversity of opinion among the reviewers. All of the reviewers raise important points, so please go through them carefully. Dr. Chary's point on how differences in SES and other demographics between interviewers and respondents impacts the perception of emotion is particularly important, and we need more details on these dynamics, as well as methods or rubrics and training techniques for interviewers which were used to measure emotional responses.

We appreciated the diverse range of reviews for this paper, and agree that they have raised some interesting points. We have responded to all comments, but have tried to strengthen these points in particular. The discussion has been significantly edited, and multiple additional references requested by the reviewers added.

Conclusion (page 12) delete the first sentence. Journal style is to avoid the phrase "first study" (see instructions to authors)

This has been removed.